

# Spawning stock recruitment creates misleading dynamics under predation release in ecosystem and multi-species models

Vidette L. McGregor[1,2], Elizabeth A. Fulton[3] and Matthew R. Dunn[1]

[1] Fisheries, National Institute of Water and Atmospheric Research Ltd, Wellington, New Zealand
[2] School of Biological Sciences, Victoria University of Wellington, Wellington, New Zealand
[3] CSIRO Oceans & Atmosphere, Hobart, TAS, Australia

## ABSTRACT

Ecosystem and multi-species models are used to understand ecosystem-wide effects of fishing, such as population expansion due to predation release, and further cascading effects. Many are based on fisheries models that focus on a single, depleted population, and may not always behave as expected in a multi-species context. The spawning stock recruitment (SSR) relationship, a curve linking the number of juvenile fish to the existing adult biomass, can produce dynamics that are counter-intuitive and change scenario outcomes. We analysed the Beverton–Holt SSR curve and found a population with low resilience when depleted becomes very productive under persistent predation release. To avoid implausible increases in biomass, we propose limiting recruitment to its unfished level. This allows for specification of resilience when a population is depleted, without sudden and excessive increase when the population expands. We demonstrate this dynamic and solution within an end-to-end ecosystem model, focusing on myctophids under fishing-induced predation release. We present one possible solution, but the specification of stock-recruitment models should continue to be a topic of discussion amongst multi-species and ecosystem modellers and empiricists going forward.

## INTRODUCTION

The advent of multi-species and ecosystem models is beginning to extend and augment the fisheries management advice provided by single-species population models, as part of a global move towards ecosystem-based fisheries management (*Pikitch et al., 2004*; *Skern-Mauritzen et al., 2016*; *Collie et al., 2014*). A variety of ecosystem modelling approaches are now available (*Plagányi, 2007*), and whilst some are highly complex full-ecosystem (end-to-end) models, all have numerous simplifying assumptions to remain tractable, and many include components for population dynamics that were developed for single-species population models for fisheries. Examples of end-to-end models suitable for addressing broad questions include Atlantis (*Fulton et al., 2011*), EwE

Corresponding author
Vidette L. McGregor,
vidette.mcgregor@niwa.co.nz

(*Christensen & Walters, 2004*; *Pauly, Christensen & Walters, 2000*), and GADGET (*Begley & Howell, 2004*), and simpler models include multispecies size-based models (*Blanchard et al., 2014*; *Pope et al., 2006*) and models of intermediate complexity for ecosystems (MICE) (*Plagányi et al., 2014*; *Doonan et al., 2016*), the latter of which are used for providing more tactical advice.

The model relating a populations spawning stock biomass (SSB) to subsequent recruitment, the spawning stock recruitment (SSR) curve, is one component that has transferred from fisheries population models into ecosystem and multispecies models. Spawning stock is usually measured as biomass by weight, and recruitment as numbers at age 1, such that the relationship describes the number of individuals expected in recruitment for a given mature biomass. The *Beverton & Holt (1993)* is the most commonly used SSR curve (*Shertzer & Conn, 2012*), which assumes that recruitment increases with spawning stock size to an asymptote, and therefore it does implicitly assume some density-dependent ecosystem effect, including competition for resources, although the form of this effect is non-specific. While it is possible for some ecosystem models to model the larval stage in sufficient detail such that a SSR relationship is not assumed but is an emergent property (e.g. OSMOSE; *Shin & Cury, 2004*), most ecosystem and multispecies models do not include sufficient detail for this, and hence a SSR curve is still assumed.

The Beverton–Holt SSR (Eq. (1)) is defined such that when the SSB is at its unfished level ($B_0$), the number of recruits produced is $R_0$. These parameters are not the upper limit for biomass or recruitment as these are also influenced by natural mortality. The asymptote of the Beverton–Holt SSR curve is greater than $R_0$, hence if the SSB exceeds $B_0$ as it may when mortality is low, recruitment will exceed $R_0$. Once recruitment joins a population, the amount retained for future years, and hence contributing to the SSB, will also be influenced by natural mortality.

$$R = \frac{\alpha S}{\beta + S} \tag{1}$$

where

$R$ is the number of recruits

$S$ is the SSB.

When modelling a population with low resilience, the curve can be specified such that the initial part of the curve does not increase as steeply, thus when the population is depleted, it will take longer to recover. The resilience aspect will not affect the unfished state of the population providing the point ($B_0$, $R_0$) remains the same. In fisheries modelling, resilience at low biomass levels is defined using the term steepness ($h$), which is the proportion of $R_0$ recruited when SSB is 20% $B_0$ (*Mace & Doonan, 1988*; *Francis, 1992*; *Haddon, 2001*; *Lee et al., 2012*). The initial slope of the curve is steeper when defined with a high $h$ and shallower when defined with a low $h$. To preserve the populations unfished state, the curve passes through ($B_0$, $R_0$) for all values of $h$ (Fig. 1). Both the preservation of the unfished state and the specification of resilience when the population is depleted are important aspects of the SSR for fisheries modelling whether in a multi- or

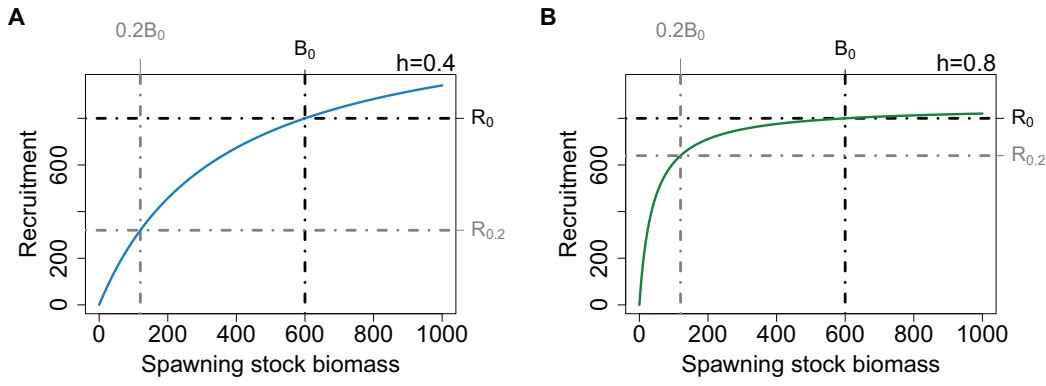

**Figure 1** Beverton–Holt SSR curve with $h = 0.4$ (A, blue line) and $h = 0.8$ (B, green line), with $B_0$ and $R_0$ (black dot-dashed lines) and 20% $B_0$ and $R_{0.2}$ (grey dot-dashed lines).

single-species context. Unfortunately, it is these two aspects that lead to counter-intuitive model behaviour when a population expands beyond its unfished state. As the curve is defined to pass through the point $(B_0, R_0)$, a slow initial increase results in a steeper increase near $(B_0, R_0)$. Hence, when a populations biomass exceeds its unfished level $(B_0)$, the effect of $h$ reverses and a population defined to have low resilience when depleted will increase more rapidly as it passes $B_0$. Changing the initial slope of the curve while keeping the recruitment asymptote unchanged, would require the virgin state of the population to change, and hence we have not explored it further.

The switching of the steepness effect has not been such a concern for fisheries assessment models due to stochasticity, statistical fitting to data, and depleted populations. When used in a stochastic population model for fisheries assessments, the SSR curve is combined with recruitment deviates that allow for between year variability. Both the deviates and unfished recruitment $(R_0)$ are estimated using statistical methods to best fit data, hence reducing the concern of implausible recruitment dynamics (*Bull et al., 2012*; *Fournier, Hampton & Sibert, 1998*; *Methot & Wetzel, 2013*). When used for projections, the situation is similar to that in multi-species and ecosystem models, except the population modelled is likely in a depleted state due to fishing and unlikely to exceed its unfished state during the model projection, typically only around 5 years.

Understanding the impacts of fishing on ecosystems requires careful specification relating to flow-on or cascading effects. Inherent in ecosystem and multispecies models is time-varying natural mortality, the sources of which may include predation, disease or starvation. When natural mortality decreases such as from predation release resulting from fishing pressure, a populations biomass may increase beyond its unfished level $(B_0)$. As we demonstrate here, it is possible for a populations biomass to increase an unrealistic amount in response to fishing pressure on its' predators, and without careful inspection of the SSR curve, the cause of this could remain a mystery. Worse still, such increases could be accepted as valid responses and erroneously affect ecosystem indicators giving misleading results to fisheries scenarios.

A simple solution may be to cap recruitment at $R_0$, such that the SSR relationship is the same for all steepness values when the population exceeds its virgin level. There may be other solutions, perhaps involving damping, switches or smoothers, but it seems a simple solution is a good place to start. There may also be other solutions using the Ricker SSR model (*Ricker, 1954*), although this model brings with it different assumptions around the SSR relationship. Similar to the Beverton–Holt curve, the Ricker model can be parameterised in terms of steepness, and the effect of steepness switches when biomass increases beyond $B_0$, but it also has additional complexity due to all curves going below $R_0$ when biomass is greater than $B_0$. It is the simple solution of capping recruitment at $R_0$ using the Beverton–Holt SSR curve, that is explored in this study. While it may not be the best solution for all situations—there may be populations that require recruitment to vary when a population is larger than its virgin level, the simplicity of this solution makes exploration of its application clearer.

One of the most abundant and widespread groups of fish this could affect in ecosystem or multispecies modelling are myctophids (also called lanternfishes). Myctophids make up roughly 65% of all deep-sea fish biomass in the oceans (*Paxton, Eschmeyer & Kirshner, 1998*), are a key prey of many commercially important fisheries species globally (*Irigoien et al., 2014*; *Koz, 1995*; *Clark, 1985*; *Collins et al., 2008*; *Young et al., 2010*) and have fairly low fecundity rates (*Catul, Gauns & Karuppasamy, 2011*). These schooling forage fish are critical to a healthy marine ecosystem both in terms of energy flows within the food web, and nutrient recycling as they act as a biological pump through their feeding-motivated daily vertical migrations (*Hernández-León et al., 2010*). As such, they may be influential in ecosystem indicators used to evaluate ecosystem-wide effects of fisheries scenarios within ecosystem and multispecies models.

In this study, we first explore the Beverton–Holt SSR curve under varying natural mortality with different steepness values using a simple population model. We then present fishing-induced predation release of myctophids using an end-to-end ecosystem model of the Chatham Rise, New Zealand (*McGregor et al., 2019*). We compare resulting biomass trajectories from the end-to-end ecosystem model of myctophids under predation release using different steepness values, with and without capping recruitment. We calculate key ecological indicators from fishing scenarios with and without capping recruitment.

## SIMPLE POPULATION MODEL WITH VARYING NATURAL MORTALITY

We explored the effect of Beverton–Holt SSR on population abundance with time-varying mortality and different steepness values by simulating a simple population (Eq. (2)). We used case study values of $B_0$ = 600 tonnes, $R_0$ = 800 thousand individuals and ran 13 simulations, each with a different $h$ from 0.35 to 0.95. Base instantaneous natural mortality rate ($M$) was calculated to produce a constant population where the number of deaths match the number of recruits and held the population at $B_0$ (Eq. (3)), with $N_0$ the product of $B_0$ and the average weight of spawning stock individuals (set at 500 g). Time-varying mortality was constructed using the sine curve to include a period of high
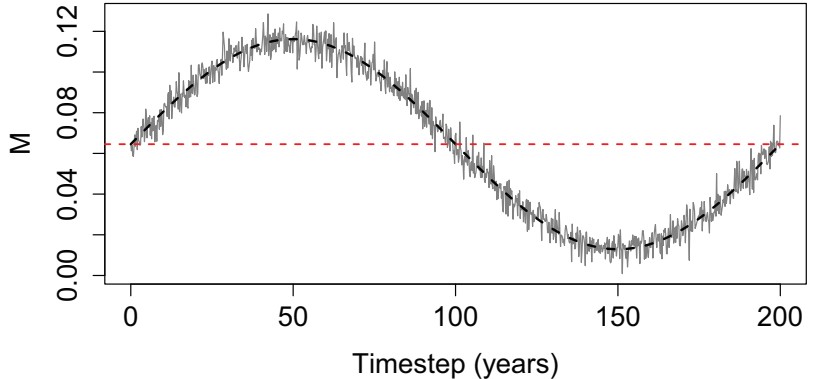

**Figure 2 Time-varying mortality ($M$) used for simple population model.** The horizontal dashed line is the base mortality (base $M$) that would result in the population staying at $B_0$.

mortality followed by a period of low mortality, and with random variability added using random variable $X \sim N(0, 0.005)$ (Eq. (4) and Fig. 2).

$$N_t = (N_{t-1} + R_{t-1})e^{-M} \tag{2}$$

where

$N_t$ is the abundance in numbers at time $t$

$R_t$ is the recruitment in numbers at time $t$

$$M = -\ln\left(\frac{N_0}{N_0 + R_0}\right) \tag{3}$$

$$M_t = 0.8\text{M} \times \sin\left(\frac{\pi t}{100}\right) + M + X \tag{4}$$

where $t$ is the timestep (year), $M$ is the base $M$,

$X \sim N(0, 0.005)$.

The effect of $h$ on the simulated SSB was as expected when $M$ was high and the population depleted; the effect was reversed when $M$ was low and the population increased beyond $B_0$ (Fig. 3). SSB trajectories were lower when SSR was parameterised with low steepness ($h$) during the period of high mortality, as is the intended effect of steepness—a population with low $h$ is less resilient and if depleted it will take longer to recover. SSB trajectories during the period of low mortality were higher when SSR was parameterised with low $h$, which is not the intended effect of steepness. Thus, a population deemed to have the least resilience when depleted is the most productive when mortality is reduced. The SSR curve coloured by $h$ shows the cause; recruitment is greater for a given SSB when SSB > $B_0$ (Fig. 3).

We applied a cap to recruitment at $R_0$ and re-simulated the same simple population for each steepness value (Fig. 4). The resulting biomass trajectories were identical to the un-capped version when mortality was high and SSB < $B_0$ as this part was not affected by the recruitment cap. During the period of low mortality, the trajectories almost converged

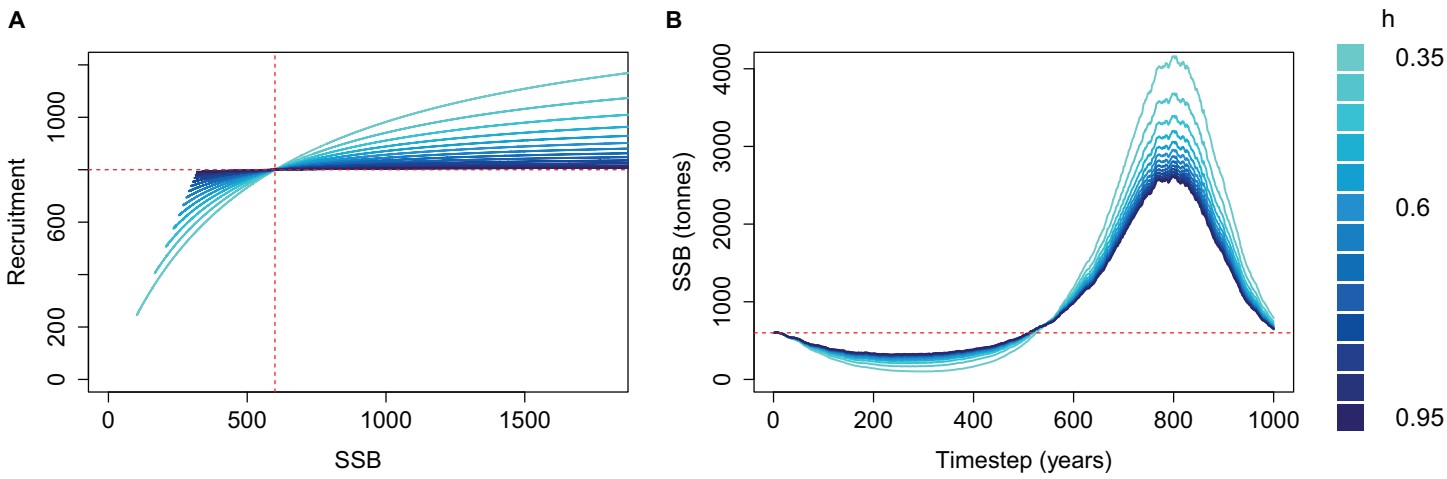

**Figure 3** Beverton–Holt spawning stock recruitment curves (A) and resulting biomass trajectories (B) from a simple population model using different steepness, *h*, ranging from 0.35 to 0.95 and time-varying mortality as in **Fig. 2**. $B_0$ shown with vertical red dash line (A) and horizontal red dashed line (B); $R_0$ shown with horizontal red dashed line (A).

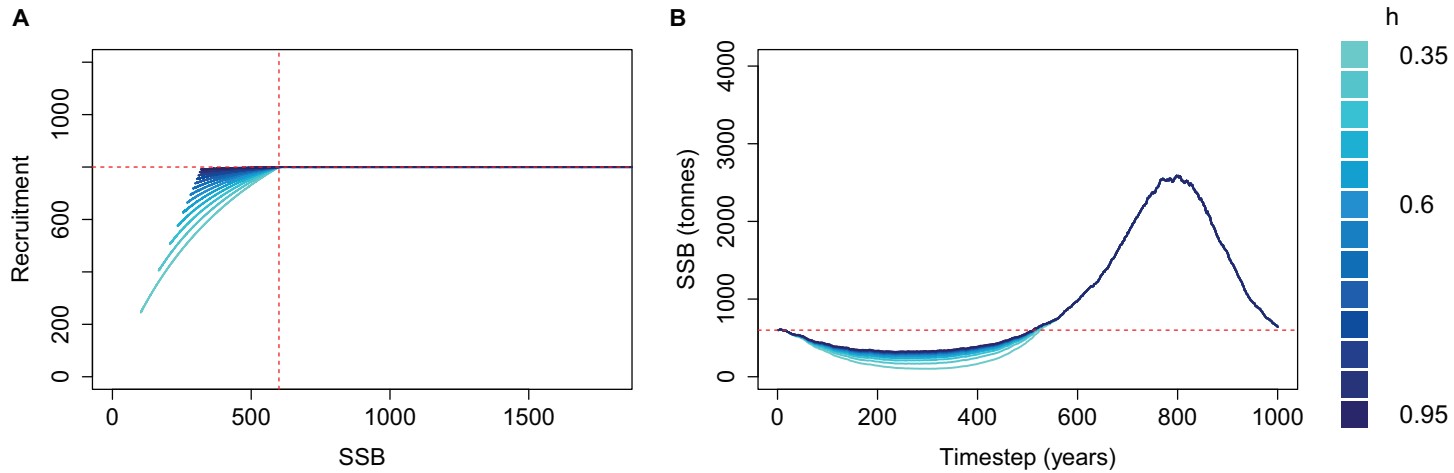

**Figure 4** Beverton–Holt spawning stock recruitment curves with recruitment capped at $R_0$ (A) and resulting biomass trajectories (B) from a simple population model using different steepness, *h*, ranging from 0.35 to 0.95 and time-varying mortality as in **Fig. 2**. $B_0$ shown with vertical red dash line (A) and horizontal red dashed line (B); $R_0$ shown with horizontal red dashed line (A).

to the same curve, rather than increasing further when using lower steepness. Hence, when applying a cap of $R_0$ to recruitment, the steepness value defined to reflect resilience when the population is depleted will be inconsequential when the population increases beyond $B_0$.

## FISHING INDUCED PREDATION RELEASE

We tested the Beverton–Holt SSR dynamics within an end-to-end ecosystem model of the Chatham Rise, New Zealand, using Atlantis (*McGregor et al., 2019*; *Fulton et al., 2011*). The Chatham Rise Atlantis model can be run with forced catches, allowing us to analyse

the resulting model estimates of biomass. This model consists of 53 species functional groups, 37 of which were modelled with age-structure, and the remainder as biomass pools. Myctophids (modelled with age-structure) were selected as the case-study species group for recruitment dynamics as they are an abundant prey species of many key fisheries species in this area, including hoki (*Macruronus novaezelandiae*), black oreo (*Allocyttus niger*), orange roughy (*Hoplostethus atlanticus*), and medium pelagic fish (primarily barracouta (*Thyrsites atun*)). Myctophids are also an abundant prey species globally (*Irigoien et al., 2014*), and have fairly low fecundity (*Catul, Gauns & Karuppasamy, 2011*) suggesting resilience when depleted may be low.

Catch histories for the Chatham Rise Atlantis Model are available as supplementary materials to *McGregor et al. (2019)*. Species with stock assessments have catch histories that were reconstructed under the review of technical fisheries working groups, including hoki (*McKenzie, 2016*), orange roughy (*Dunn & Doonan, 2018*), hake (*Horn, 2013*), and ling (*McGregor & Horn, 2015*). Catch histories for species without a stock assessment were reconstructed using commercial catch and effort data from the 'Warehou' database administered by Fisheries New Zealand (1989–present day), the Fisheries Statistics Unit database administered by NIWA (1974–1989), and from Annual Reports on Fisheries (1900–1973). While the reliability of these data varies across species and time, and the quality of data reporting likely improved with the introduction of New Zealand's Quota Management System (QMS) in 1986, we are satisfied these catch histories are a sufficiently accurate representation both in trend and magnitude.

We simulated the Chatham Rise Atlantis model with alternate SSR specification for myctophids consisting of all combinations of steepness $h \in \{0.5, 0.6, 0.7, 0.8, 0.9\}$ and recruitment capped at $R_0$ or not—10 model simulations in total. From the mid-1970s when many of the Chatham Rise fisheries became established (Fig. 5), the effects of predation release on myctophids in the model become apparent (Fig. 6). Their biomass increased beyond $B_0$ and increased more with lower recruitment steepness ($h$), with low $h$ producing the greatest resulting biomass. When we capped recruitment at $R_0$, the biomass was still able to increase as we may expect under predation release, but the increase was the same regardless of $h$. This allows for the specification of a species that is less resilient when depleted without it increasing more rapidly when released. This was similar to the low-mortality response in the simple population model of the previous section in which we controlled time-varying natural mortality. The Atlantis model simulations gave a brief period of high mortality (near 1980), and this only affected the biomass when steepness was lowest and recruitment was capped at $R_0$ (yellow line, right plot, Fig. 6). When recruitment was not capped, the low steepness simulation kept biomass above $B_0$ when the capped version was not able to do this.

## ECOLOGICAL INDICATORS

We evaluated a range of ecological indicators intended to assess the effect of fishing on an ecosystem and tested for sensitivity in the response to recruitment steepness of myctophids and whether their recruitment was capped at $R_0$, using the outputs from the Chatham Rise Atlantis model, including estimated biomass, numbers-at-age, and size-at-age.
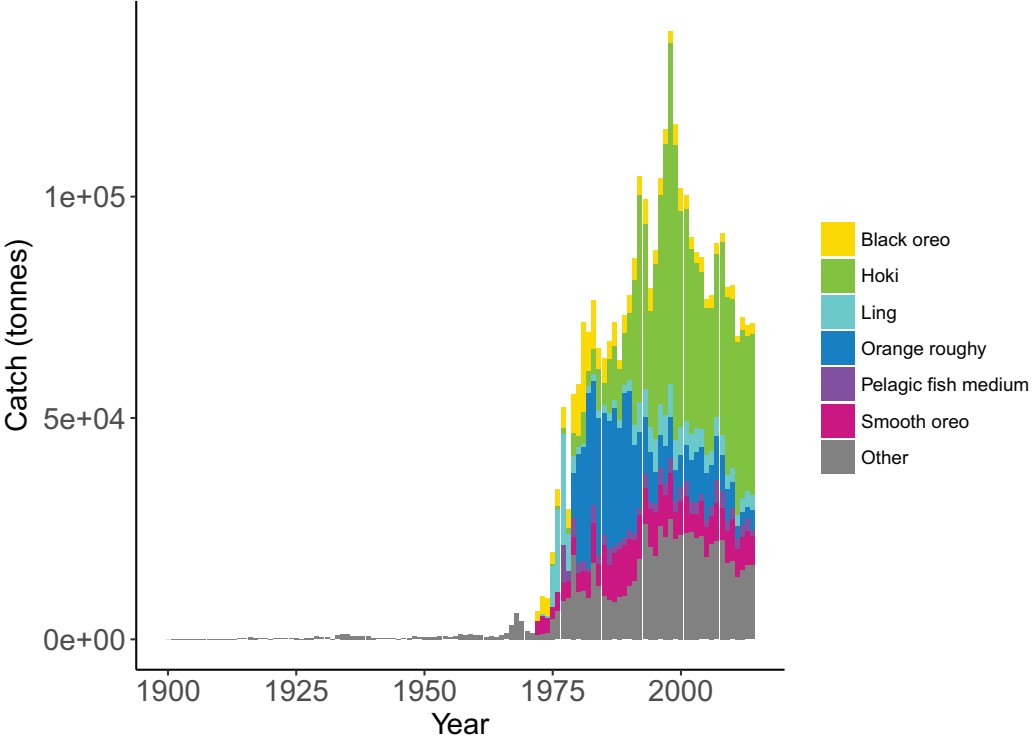

**Figure 5 Historical catches from 1900 to 2014 with top six species groups based on total catch coloured separately (*McGregor et al., 2019*).**

The ecological indicators selected (descriptions and references in Table 1) were those that have gained traction in the literature and were suited to the Chatham Rise (*Tuck, Cole & Devine, 2009*). Kempton's species diversity index measures the number of species relative to the total biomass. The adaption by *Ainsworth & Pitcher (2006)* uses species groups rather than individual species and the 10th percentiles to eliminate any very small or large species groups in terms of biomass (Eq. (5)). All indicators were calculated for the hindcast period (1900–2016) for six models consisting of three steepness values (0.5, 0.7, 0.9) and either capped or not capped recruitment for myctophids, and for projections using all of these models under three catch scenarios (zero catch, *status quo* catch, half catch), giving 18 projections. *Status quo* catch was taken to be the average from the most recent 5 years (2012–2016) for each fished species group, and half catch was half the *status quo* catch.

$$Q = \frac{0.8\psi}{\log(\rho_2/\rho_1)} \tag{5}$$

$Q$ is the adapted Kempton's species diversity index from *Ainsworth & Pitcher (2006)*
$\psi$ is total number of functional species groups
$\rho_1$, $\rho_2$ are the lower and upper 10th percentiles in the cumulative abundance distribution.

All indicators except landings/primary productivity had different responses to both recruitment steepness and recruitment cap in either or both the hindcast simulations and

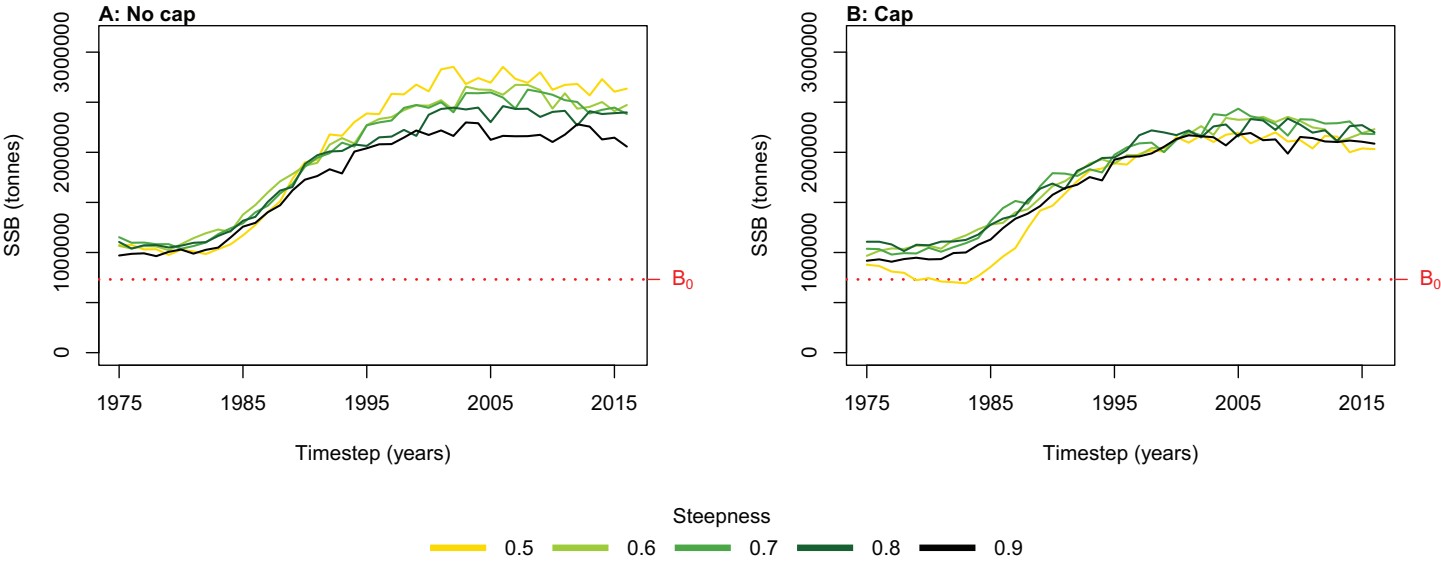

**Figure 6 Hindcast biomass trajectory for the myctophids species functional group with different recruitment steepness ($h$) and no cap on recruitment (A) and with recruitment capped at $R_0$ (B).**

**Table 1 Ecological indicators evaluated for hindcast simulations and catch scenario projections, with key references for each indicator.**

| Indicator | Reference |
|---|---|
| Kempton's species diversity index | *Ainsworth & Pitcher (2006)* |
| Mean trophic level | *Pauly & Watson (2005)* and *Shin et al. (2018)* |
| Biomass/landings | *Shin et al. (2010, 2018)* |
| Landings/biomass of primary producers | *Link (2005)* |
| Biomass of pelagic fishes/biomass total | *Link (2005)* |
| Biomass trophic level 4 and above/biomass trophic level 3 | *Link (2005)* |

**Table 2 Sensitivity (indicated by ✓) of ecological indicators to steepness and recruitment cap in the hindcast and/or catch scenario projections.**

| Indicator | Steepness | Recruitment cap |
|---|---|---|
| Kempton's species diversity index | ✓ | ✓ |
| Mean trophic index | ✓ | ✓ |
| Biomass/catch | ✓ | ✓ |
| Landings/biomass of primary producers | – | – |
| Biomass of pelagic fishes/biomass total | ✓ | ✓ |
| Biomass trophic level 4 and above/biomass trophic level 3 | ✓ | ✓ |

the catch scenario projections (summarised as 'sensitivity' in Table 2). These indicators may help understand the effect of fishing on the ecosystem—whether the effect is positive, negative, or perhaps caused a shift in the system. They focus on different aspects of the system, and hence may give different impressions. Imposing a cap on recruitment for

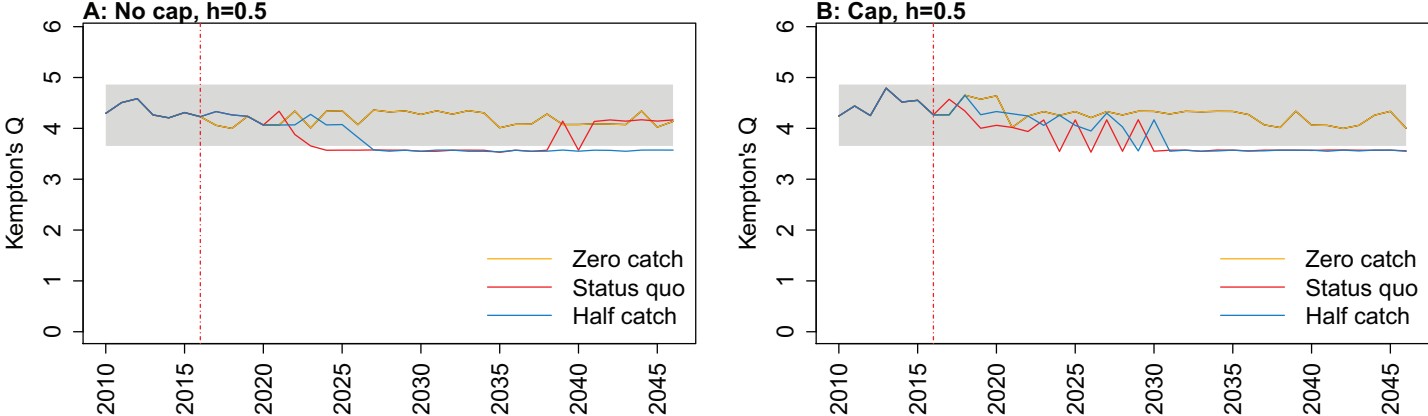

**Figure 7** Kempton's Q calculated from Chatham Rise Atlantis model simulations with recruitment steepness set at 0.5 for myctophids, no cap on recruitment (A), recruitment capped at $R_0$ (B), and three catch scenarios: (1) Zero catch; (2) *Status quo* catch; (3) Half catch, for the 2010–2016 hindcast period and 2016–2046 projection period. The red dashed vertical line marks the last hindcast year, 2016. The grey region shows the range of values from the period 1900 to 1970 when fishing was non-existent or very small.

myctophids generally suggested more favourable responses of the ecosystem to fishing, except for biomass over catch which was reduced. The full set of figures for ecosystem indicators are in Appendices A–E, with subsets in the text (Figs. 7–11) that most effectively illustrate the analyses.

Kempton's Q, a measure of diversity as adapted for ecosystem models by *Ainsworth & Pitcher (2006)*, suggested the models with no cap on myctophid recruitment were more likely to suffer a drop in diversity earlier (Fig. 7; Fig. A1). The models with a cap on myctophid recruitment indicated the system would not recover in terms of diversity within the next 30 years in the *status quo* and half catch scenarios, but the uncapped models recovered providing fishing was at the *status quo*. In the un-capped models, recovery occurred earlier when steepness was lower. It seems counter-intuitive the *status quo* catch scenarios would recover in terms of diversity and the reduced catch scenario (half catch) would not. Given the higher catch scenario brings recovery and the timing is earlier for lower myctophid steepness, this is likely due to the additional myctophids in the system due to predation release and the reversed effect of steepness.

Mean trophic level was calculated for the age-structured species groups in the model. This was affected by steepness in the hindcast models with no cap on recruitment and the corresponding projections, although the effect was not linear. Mean trophic level was lowest when myctophids had highest steepness (0.9) and there was little difference between steepness of 0.5 and 0.7, although the latter was slightly higher (Fig. 8; Fig. A2). There is some feedback within the model with medium and large pelagic fishes increasing in biomass with myctophids, and this is likely to come through in this trophic level indicator. All the models with capped recruitment gave higher mean trophic levels both in the hindcast and projections.

Biomass over catch was calculated using the biomass of age-structured species groups as this covers the species that may be fished. It was affected by recruitment steepness when recruitment was not capped, although the effect was fairly subtle, and perhaps

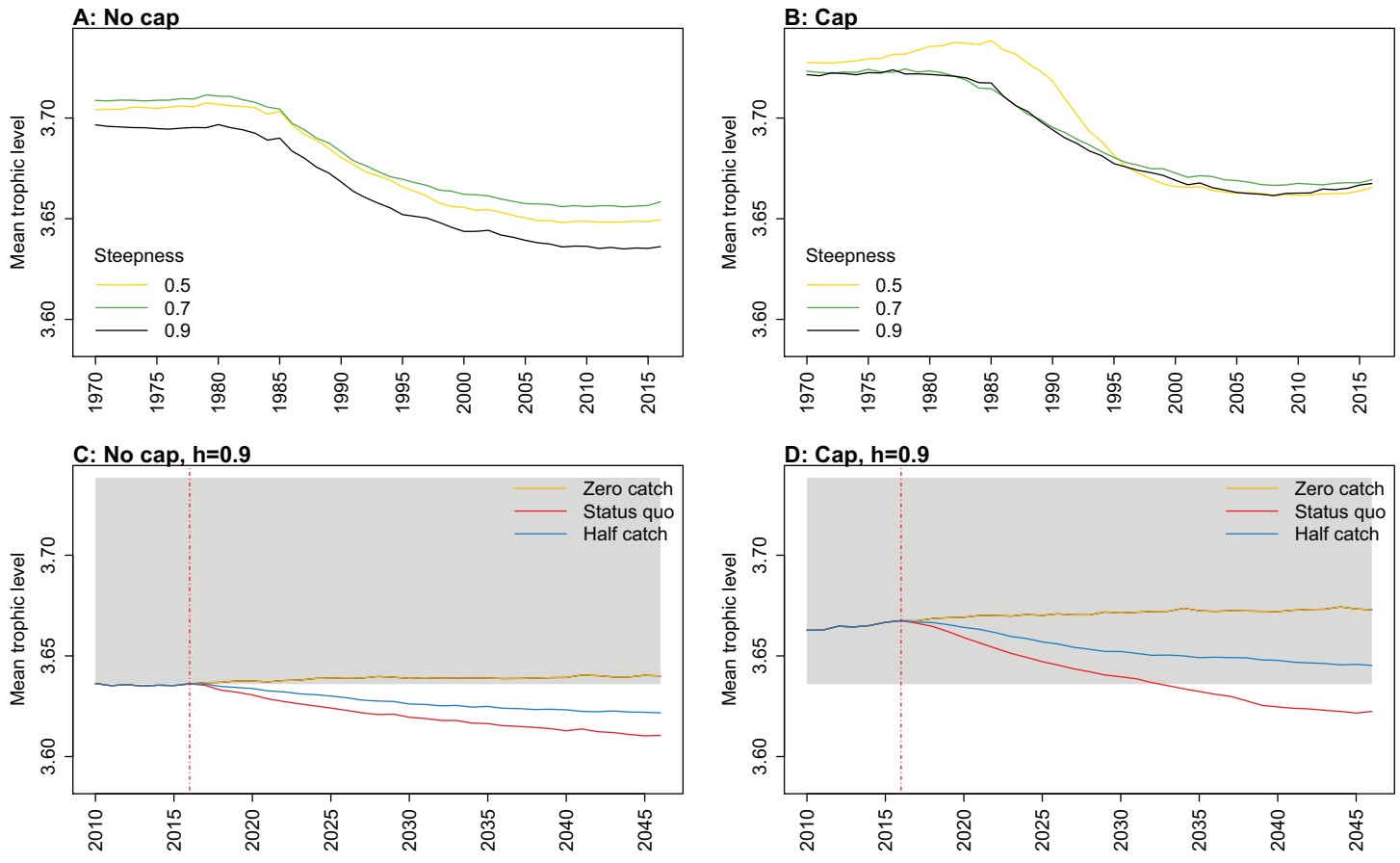

**Figure 8 Mean trophic level of age-structured species groups calculated from Chatham Rise Atlantis model simulations with no cap on recruitment (A, C), recruitment capped at $R_0$ (B, D), recruitment steepness values $h \in (0.5, 0.7, 0.9)$ for the 1970–2016 hindcast period (A, B) and $h$ set at 0.9 for myctophids, with three catch scenarios: (1) Zero catch; (2) *Status quo* catch; (3) Half catch, for the 2016–2046 projection period (C, D).** The red dashed vertical line marks the last hindcast year, 2016. The grey region shows the range of values from the period 1900 to 1970 when fishing was non-existent or very small. 

dwarfed by the large and sudden increase in this ratio when catches were halved (Fig. 9; Fig. A3). In all cases, halving the catches resulted in the biomass/catch ratio increasing suddenly, as expected. The ratio was highest when recruitment was not capped, and steepness was lowest. While this pattern was present in all hindcast and projections, it was most evident when steepness was 0.5 and recruitment was not capped. When recruitment was capped, there was a dip in the ratio in the 1980s–mid-1990s in the low-steepness model. This reduction due to low steepness is a more plausible effect as we would expect there to be less myctophids when they are defined with low steepness. As we have not included biomass-pool species groups in this index, the main prey groups of myctophids were excluded. Hence, a reduction in myctophids was not offset in the by an increase in their prey.

While biomass of pelagic fishes over total biomass exceeded previous levels under *status quo* fishing for all recruitment scenarios, it was highest when steepness was lowest (0.5) and recruitment was not capped (Fig. 10; Fig. A4). For this recruitment scenario,

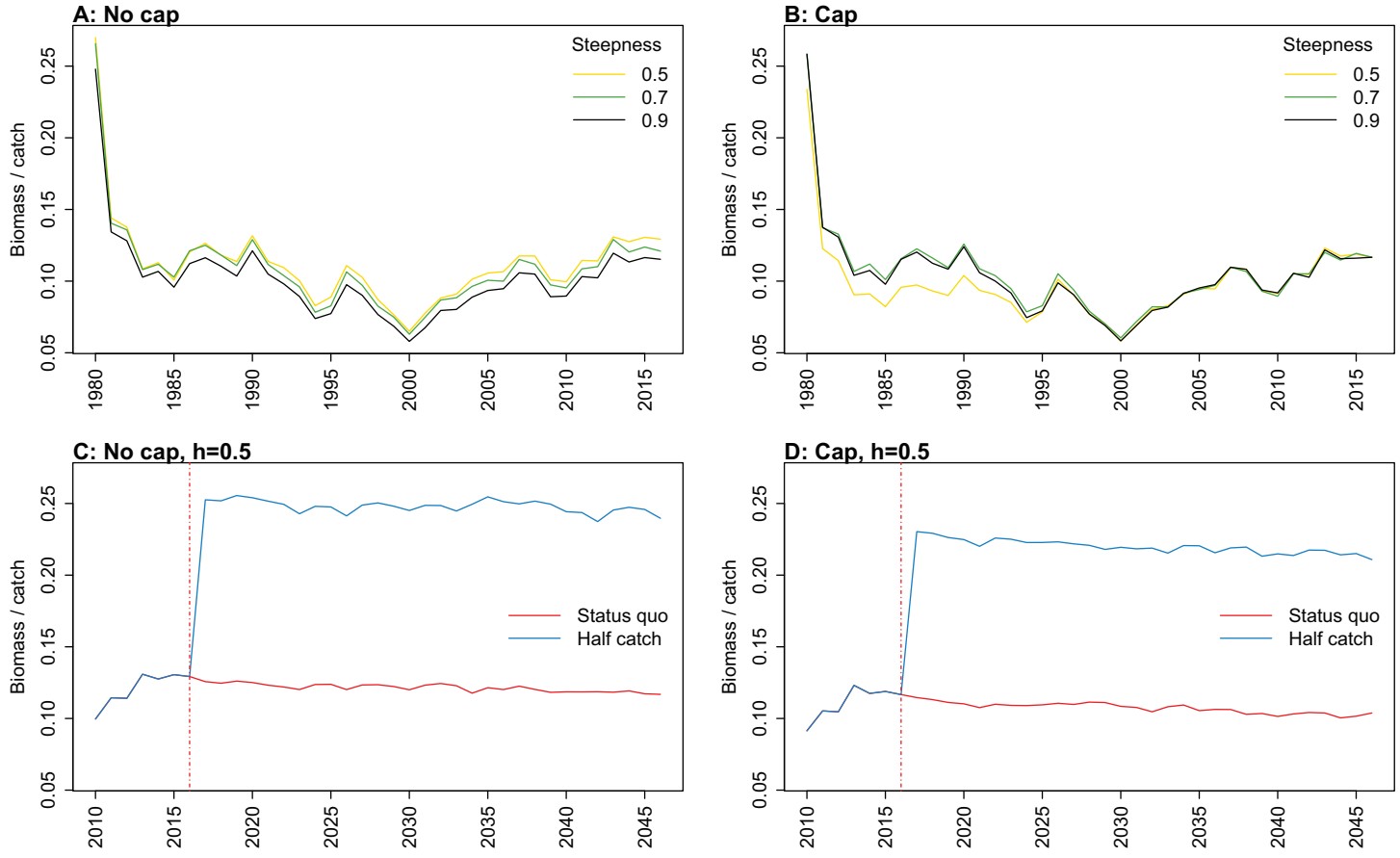

**Figure 9** Biomass of age-structured species groups over catch calculated from Chatham Rise Atlantis model simulations with no cap on recruitment (A, C), recruitment capped at $R_0$ (B, D), recruitment steepness values $h \in (0.5, 0.7, 0.9)$ for the 1970–2016 hindcast period (B, D) and $h$ set at 0.9 for myctophids, with three catch scenarios: (1) Zero catch; (2) *Status quo* catch; (3) Half catch, for the 2016–2046 projection period (C, D). The red dashed vertical line marks the last hindcast year, 2016. The grey region shows the range of values from the period 1900 to 1970 when fishing was non-existent or very small.

the ratio for both *status quo* and half catch projections went above the previous highest level by approximately 20%. This is a direct response to the higher myctophid biomass as the low-steepness recruitment model expands under predation release.

Ratio of biomass at trophic level 4 and higher over biomass at trophic level 3 was sensitive to recruitment cap and steepness in the projections. The ratio was reduced when steepness was lower, and recruitment was not capped. In all *status quo* projections with no recruitment cap, the ratio went below the previous lowest level, and was lower when steepness was lower (Fig. 11; Fig. A5). In all projections with recruitment capped, the ratio remained within pre-fishing bounds. The warning and limit reference points for this indicator as defined by *Link (2005)* were concerned with the ratio becoming too high, with the suggested control rule to alleviate fishing on trophic level 3 species. As fishing pressure on the Chatham Rise is high on trophic level 4 species and level 3 species, this ratio becoming too low may also be of interest or concern as it relates to the balance of the ecosystem.
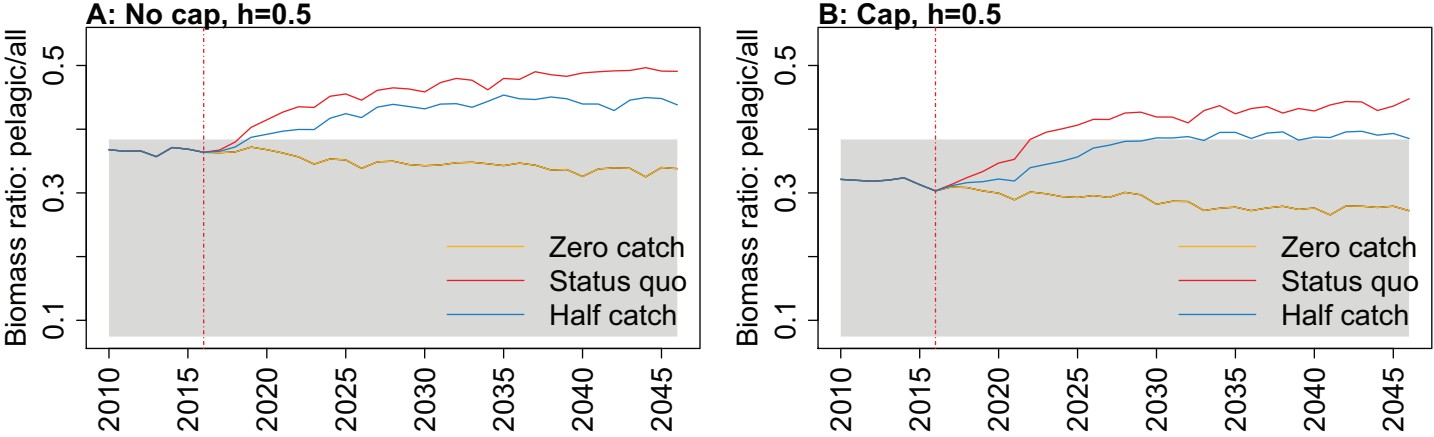

**Figure 10 Biomass of pelagic fishes over biomass of all age-structured species groups from Chatham Rise Atlantis model simulations with recruitment steepness set at 0.5 for myctophids, no cap on recruitment (A), recruitment capped at $R_0$ (B), and three catch scenarios: (1) Zero catch; (2) *Status quo* catch; (3) Half catch, for the 2010–2016 hindcast period and 2016–2046 projection period.** The red dashed vertical line marks the last hindcast year, 2016. The grey region shows the range of values from the period 1900 to 1970 when fishing was non-existent or very small.

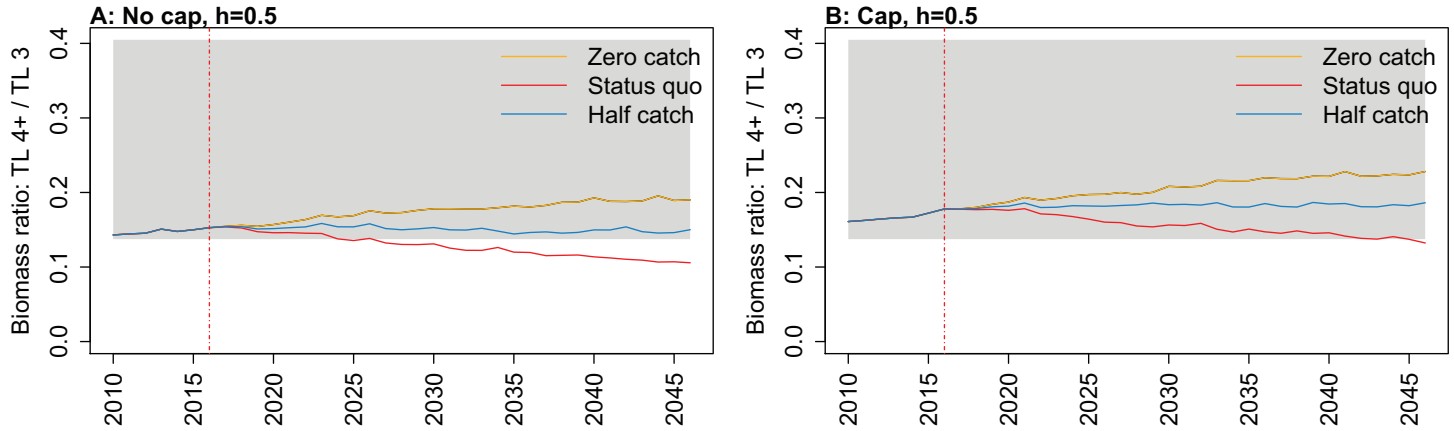

**Figure 11 Biomass ratio of trophic level 4 and higher over trophic level 3 from Chatham Rise Atlantis model simulations with recruitment steepness set at 0.5 for myctophids, no cap on recruitment (A), recruitment capped at $R_0$ (B), and three catch scenarios: (1) Zero catch; (2) *Status quo* catch; (3) Half catch, for the 2010–2016 hindcast period and 2016–2046 projection period.** The red dashed vertical line marks the last hindcast year, 2016. The grey region shows the range of values from the period 1900 to 1970 when fishing was non-existent or very small.

## DISCUSSION

Spawning stock recruitment parameter 'steepness' that reflects a population's resilience when depleted, is an influential and often unknown parameter in fisheries stock assessment models and is often used in ecosystem and multispecies models (*Lee et al., 2012*). Regardless of whether we define resilience using the fisheries term steepness or not, this attribute of the SSR curve still exists and is important for specifying the dynamics of a modelled population. It is important to be able to define the slope of the initial increase of the recruitment curve without changing the level of recruitment expected when there is no fishing ($R_0$).

We found the effect of changing steepness in the Beverton–Holt SSR model is reversed when SSB is greater than the unfished biomass. This matters when modelling populations where mortality may be reduced such as from predation release, as this has the potential to allow a population to increase beyond its unfished level. Using the Beverton–Holt SSR model, a highly productive population with a high steepness value would become less productive when SSB exceeds unfished biomass than a population that has a low steepness value, even though both are defined with the same unfished recruitment and biomass.

When recruitment is capped at its unfished level, a population may still increase beyond its unfished state under reduced mortality, but it will not increase more with lower steepness. Hence, steepness will only affect a population when it is depleted. This is an improvement on the effect of steepness being reversed, but there may be an alternative that allows for a population with low steepness to continue to be less productive, for situations when this dynamic is appropriate.

We explored the ecosystem wide responses to fishing with varying steepness and capping or not capping recruitment for the myctophids species group. This group are abundant in marine ecosystems including the Chatham Rise, and they are prey to many key fisheries species. Even though we were only changing recruitment in one species group, this did result in changes in ecosystem indicators. Scenarios modelled without a cap on myctophid recruitment were more likely to suggest a shift in the ecosystem through increased biomass of pelagic fishes with respect to the total biomass, and more likely to suggest the systems diversity would recover if fishing remained high. Scenarios modelled with a cap on myctophid recruitment were less likely to suggest reduced average trophic level in the system in response to fishing, and the diversity of the system took longer to decline.

The results of ecosystem indicators with respect to positive or negative effects of fishing do not tell us much about whether recruitment should be capped or not, but they do flag some situations that may be in danger of misleading results if we do not. The shift in the system due to higher than previously before pelagic species is one example of potentially misleading results. It does not make sense for the expansion of pelagic fishes to be more extreme when they are specified as less resilient. The recovery of diversity under the highest catch scenario when recruitment was not capped and not in the lower catch scenario, with earlier recovery for lower steepness, is also suspicious.

The decision of whether to cap recruitment or not—or to apply a different solution, may be different across models, population dynamics, and perhaps intended model use. There may be information in data or the literature to inform reasonable or likely population growth in such situations as predation release, in which case these may be used to inform this decision. If there is nothing available as a guide, then sensitivity analyses are a sensible option, at least to ascertain the possible responses of the modelled system to an explored scenario. This approach is not unique to the specification of the SSR curve; it holds for all model development and exploration. It is the responsibility of the modeller to understand the dynamics of the model they are creating, not just from comparing biomass trends to observations, but actually understanding the dynamics relating to flow-on effects, and the implications that come with every parameterisation and

specification of their model. Where there are uncertainties in the model, results from explored scenarios need to reflect these.

There is traction in the literature for controlling fishing effort on different trophic levels such as through balanced harvesting (*Jacobsen, Gislason & Andersen, 2014*; *Reid et al., 2016*, and critiqued in *Froese et al., 2015*). Key to this idea is the response of prey species under predation release, as it aims to replace this predation mortality with fishing mortality, thus taking more total catch with little negative effect to the ecosystem. An example is *Smith et al. (2011)* where comparisons are drawn between scenarios in which lower trophic levels are not fished and those where they are. As we have shown in this study, such comparisons may be misleading. Careful consideration needs to be taken for the population dynamics under scenarios involving predation release.

Key to this work is that natural mortality varies with respect to time in ecosystem and multispecies models, and this may change how we think of 'unfished' biomass and recruitment. While we can adjust the SSR model to cope with increased biomass beyond its unfished level, it does raise the question 'should we?' Perhaps when biomass increases beyond unfished biomass, $B_0$ and $R_0$ should change, and hence the specification of the SSR curve should also change. But would this additional dynamic complexity help a model in its ability to fit to or replicate historical data, and predict future trends? There are times in single species fisheries models where natural mortality varies with time and the SSR relationship remains fixed (*Nielsen & Berg, 2014*; *Deroba & Schueller, 2013*; *Johnson et al., 2014*). The species in focus is most likely kept below its unfished level due to fishing mortality, so the steepness effect for biomass greater than unfished biomass is irrelevant. However, the meaning of unfished biomass and recruitment perhaps should be reconsidered.

While some ecosystem models (e.g. OSMOSE; *Shin & Cury, 2004*) do not require a SSR curve, as the relationship is emergent due to the explicit handling of larval dynamics in the model, these models are in the minority. Moreover, models sufficiently complex to have emergent relationships will not always be appropriate, particularly when smaller or minimum realistic models are desired—for example for rapid or tactical assessments. There are strong arguments for using the simplest model sufficient for its purpose, and this extends even to ecosystem modelling and has seen the growth in popularity of approaches such as MICE (*Plagányi et al., 2014*). This means that for the foreseeable future it is likely we will remain in need of SSR models with consistent dynamics under varying natural mortality and recruitment resilience. We propose one such variant here, but it should continue to be a topic of discussion amongst ecosystem modellers and empiricists going forward.

# APPENDIX

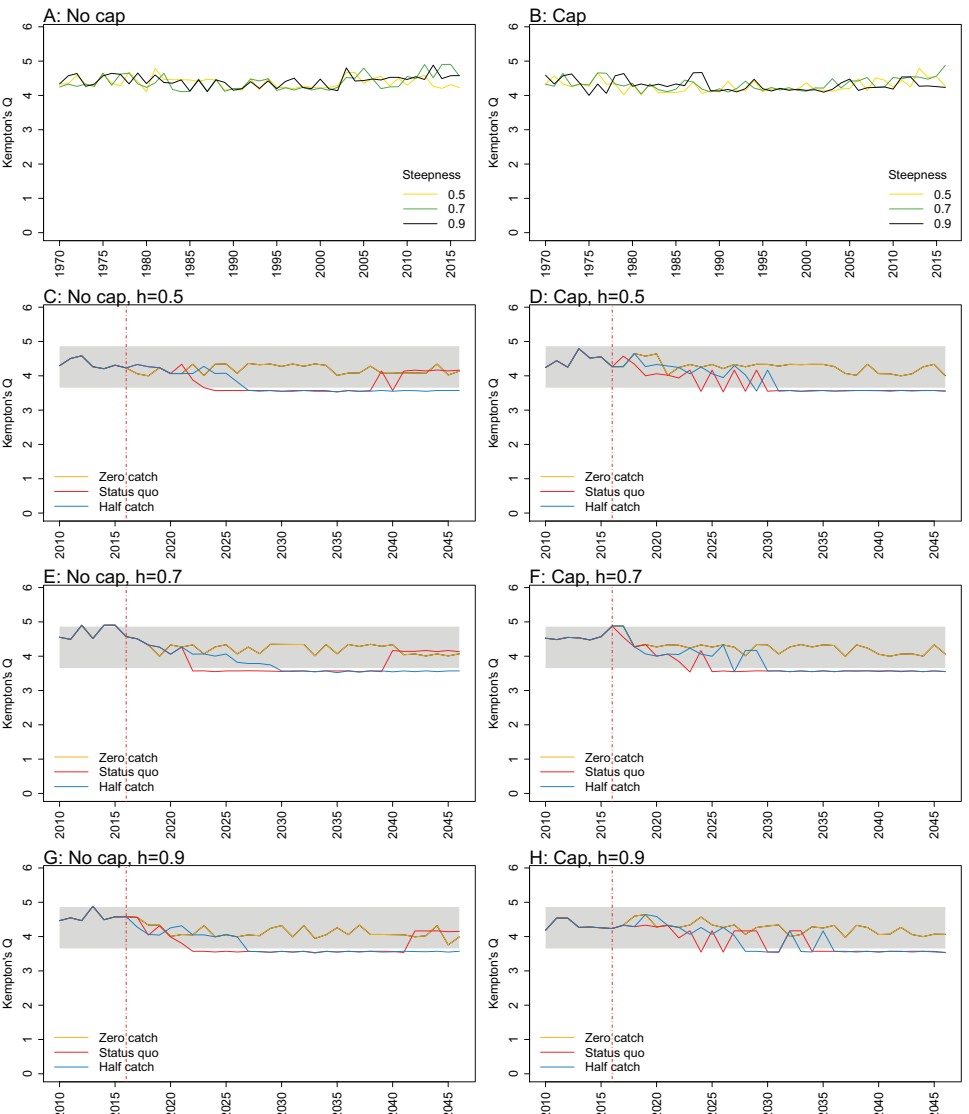

**Figure A1 Kempton's Q.** Kempton's Q calculated from Chatham Rise Atlantis model simulations with no cap on recruitment (A, C, E, and G), recruitment capped at $R_0$ (B, D, F, and H), recruitment steepness values $h \in (0.5, 0.7, 0.9)$ for the 1970–2016 hindcast period (A, B) and $h$ set at 0.5 (C, D), 0.7 (E, F), and 0.9 (G, H) for myctophids, with three catch scenarios: (1) Zero catch; (2) *Status quo* catch; (3) Half catch, for the 2016–2046 projection period (G, H). The red dashed vertical line marks the last hindcast year, 2016. The grey region shows the range of values from the period 1900–1970.

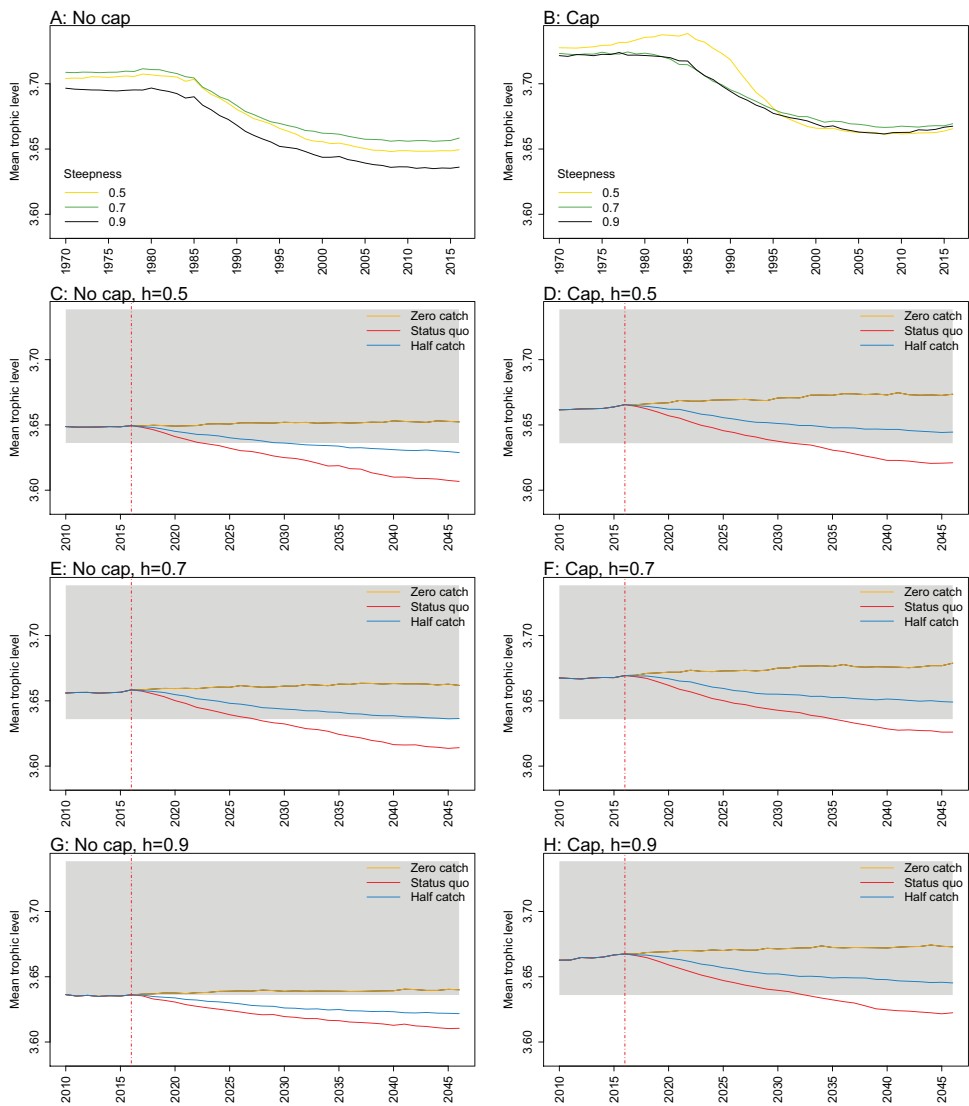

**Figure A2 Mean trophic level.** Mean trophic level of age-structured species groups calculated from Chatham Rise Atlantis model simulations with no cap on recruitment (A, C, E, and G), recruitment capped at $R_0$ (B, D, F, and H), recruitment steepness values $h \in (0.5, 0.7, 0.9)$ for the 1970–2016 hindcast period (A, B) and $h$ set at 0.5 (C, D), 0.7 (E, F), and 0.9 (G, H) for myctophids, with three catch scenarios: (1) Zero catch; (2) *Status quo* catch; (3) Half catch, for the 2016–2046 projection period (G, H). The red dashed vertical line marks the last hindcast year, 2016. The grey region shows the range of values from the period 1900 to 1970.

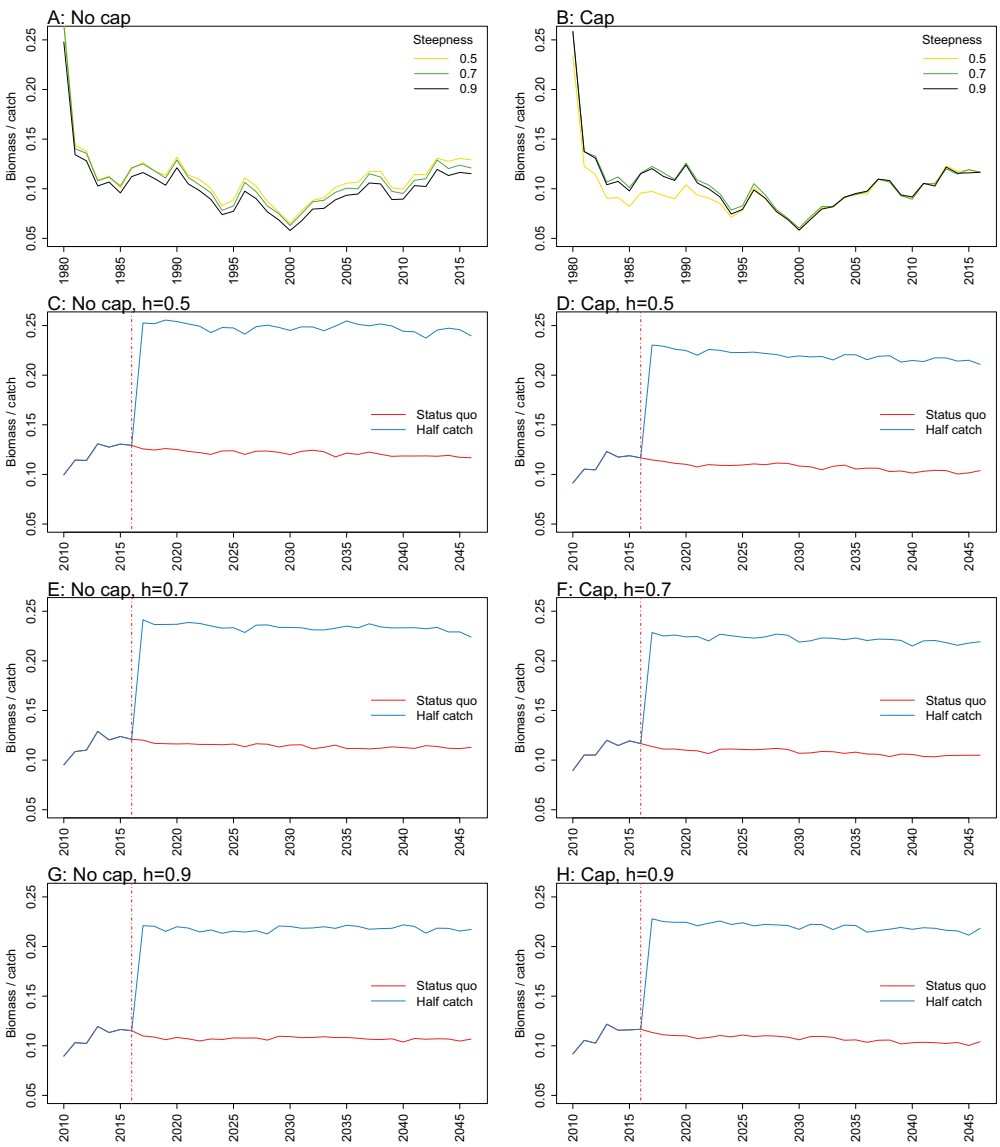

**Figure A3 Biomass over catch.** Biomass of age-structured species groups over catch calculated from Chatham Rise Atlantis model simulations with no cap on recruitment (A, C, E, and G), recruitment capped at $R_0$ (B, D, F, and H), recruitment steepness values $h \in (0.5, 0.7, 0.9)$ for the 1970–2016 hindcast period (A, B) and $h$ set at 0.5 (C, D), 0.7 (E, F), and 0.9 (G, H) for myctophids, with three catch scenarios: (1) Zero catch; (2) *Status quo* catch; (3) Half catch, for the 2016–2046 projection period (G, H). The red dashed vertical line marks the last hindcast year, 2016. The grey region shows the range of values from the period 1900 to 1970.      

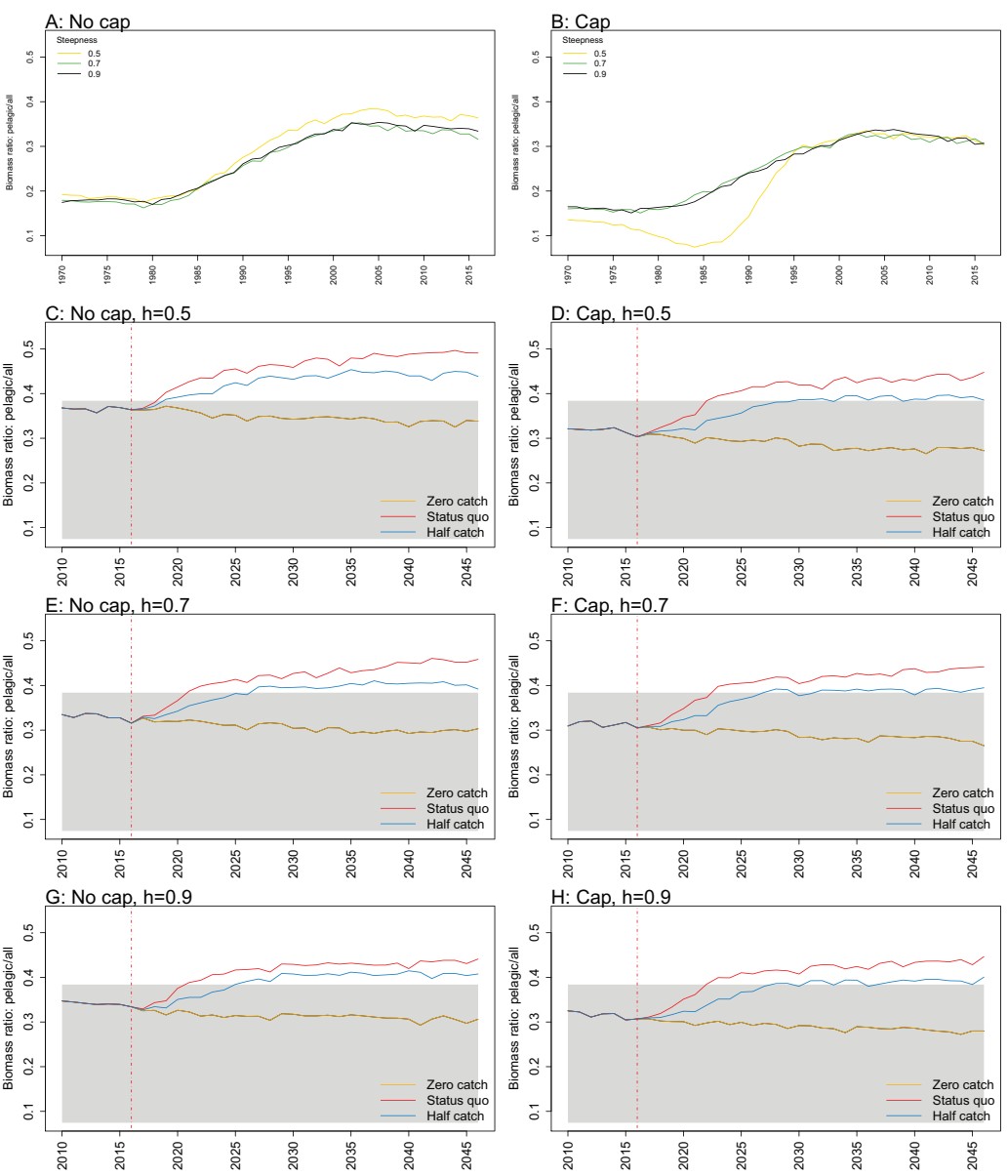

**Figure A4 Biomass pelagic over all.** Biomass of pelagic fishes over biomass of all age-structured species groups calculated from Chatham Rise Atlantis model simulations with no cap on recruitment (A, C, E, and G), recruitment capped at $R_0$ (B, D, F, and H), recruitment steepness values $h \in (0.5, 0.7, 0.9)$ for the 1970–2016 hindcast period (A, B) and $h$ set at 0.5 (C, D), 0.7 (E, F), and 0.9 (G, H) for myctophids, with three catch scenarios: (1) Zero catch; (2) *Status quo* catch; (3) Half catch, for the 2016–2046 projection period (G, H). The red dashed vertical line marks the last hindcast year, 2016. The grey region shows the range of values from the period 1900 to 1970.

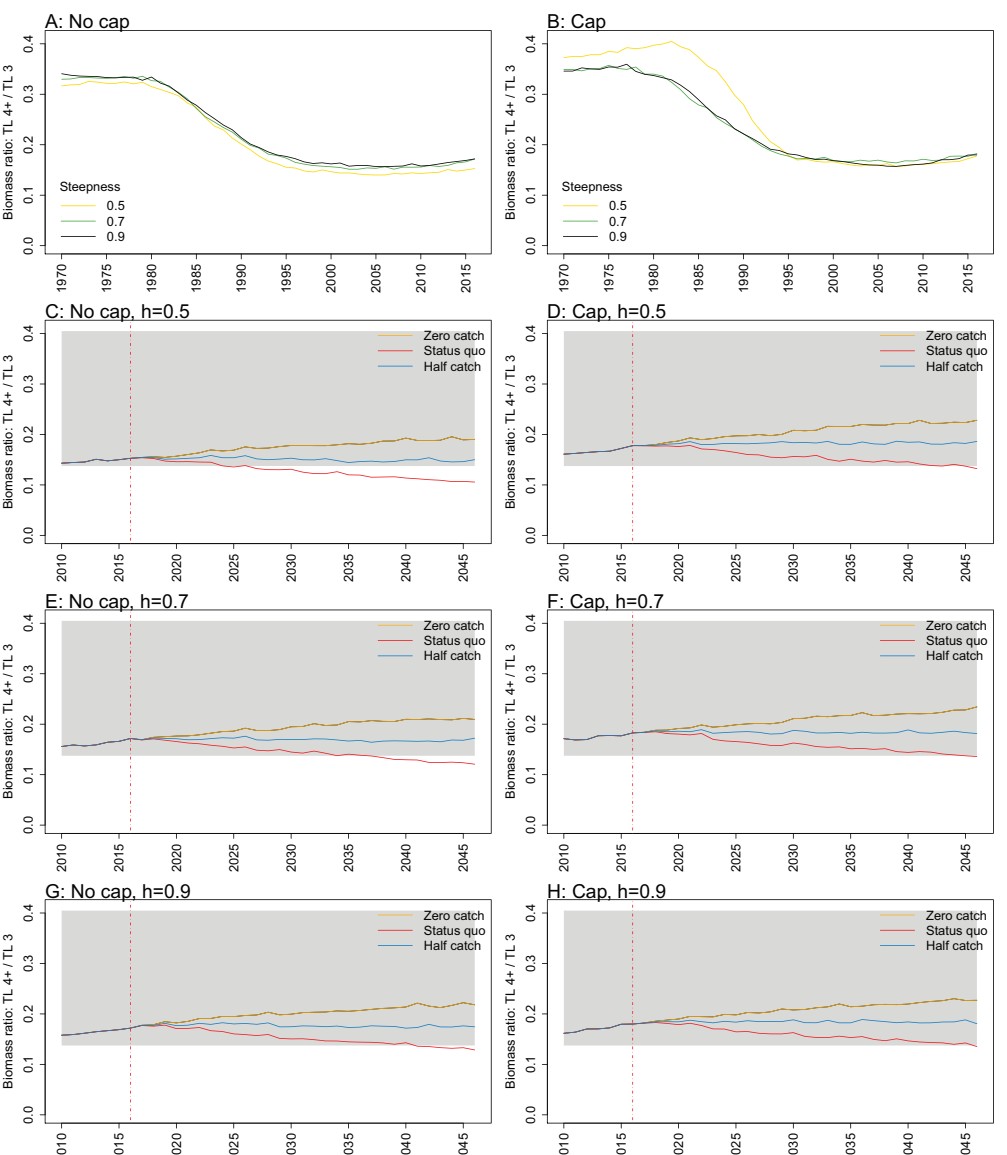

**Figure A5 Biomass trophic level 4 and higher over trophic level 3.** Biomass ratio of trophic level 4 and higher over trophic level 3 calculated from Chatham Rise Atlantis model simulations with no cap on recruitment (A, C, E, and G), recruitment capped at $R_0$ (B, D, F, and H), recruitment steepness values $h \in (0.5, 0.7, 0.9)$ for the 1970–2016 hindcast period (A, B) and $h$ set at 0.5 (C, D), 0.7 (E, F), and 0.9 (G, H) for myctophids, with three catch scenarios: (1) Zero catch; (2) *Status quo* catch; (3) Half catch, for the 2016–2046 projection period (G, H). The red dashed vertical line marks the last hindcast year, 2016. The grey region shows the range of values from the period 1900 to 1970.

## ACKNOWLEDGEMENTS

Ian Doonan, Andy McKenzie, Kath Large (NIWA), and Malcolm Haddon (CSIRO) for discussing some of the ideas, Chris Francis (independent) for reviewing the initial draft. Ian Tuck, NIWA project leader. James Bell, Victoria University supervisor.

### Funding

This work was funded under NIWA project FIFI1801. The funders had no role in study design, data collection and analysis, decision to publish, or preparation of the manuscript.

### Grant Disclosures

The following grant information was disclosed by the authors:
NIWA project: FIFI1801.

### Competing Interests

Vidette L. McGregor and Matthew R. Dunn are employed by National Institute of Water and Atmospheric Research (NIWA) Ltd. Elizabeth A. Fulton is employed by the Commonwealth Scientific and Industrial Research Organisation (CSIRO).

### Author Contributions

- Vidette L. McGregor conceived and designed the experiments, performed the experiments, analysed the data, contributed reagents/materials/analysis tools, prepared figures and/or tables, authored or reviewed drafts of the paper, approved the final draft.
- Elizabeth A. Fulton contributed reagents/materials/analysis tools, authored or reviewed drafts of the paper, approved the final draft.
- Matthew R. Dunn authored or reviewed drafts of the paper, approved the final draft.

### Data Availability

   The raw data and code is available at GitHub: https://github.com/mcgregorv/SSRmultiSpecies.

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
