# Peer review of "Spawning stock recruitment creates misleading dynamics under predation release in ecosystem and multi-species models"

_PeerJ, doi:10.7717/peerj.7308_

## Round 0.1 · original submission · Major Revisions

There is no conceptual concern regarding the manuscript. However, the following points particularly need to be addressed before considering the manuscript for publication:

- Details regarding the method and data selection must be provided in detail and clearly defined.

- Improve the content of the discussion section as suggested by reviewer #2.

Reviewer 1 ·

Basic reporting

The problem is introduced and then a potential solution is assumed in the final paragraph describing the study design. More exploration of the solution (the recruitment cap) prior to that final paragraph in the introduction would help guide the reader, perhaps discussing why they thought this was a good solution and possible pitfalls it may have. The authors make it clear in the discussion that this is only one possible solution, but more discussion of alternatives (a Ricker model?) or possible downsides (this isn’t very mechanistic?), without formally testing anything, would make the paper more complete.

The equations should be presented within the flow of the paper. “For example, Base instantaneous natural mortality (M) was calculated as: [insert equation here].”

Equation 3: While I understand S and R maybe be standard notation for the species diversity index, it is confusing in the context of this paper because a reader might think S and R are spawning biomass and recruitment.

In general, the manuscript needed copy editing, particularly with respect to use of possessive apostrophes. Could copy and paste into a word processor to check for errors? I also found the writing to be unclear at times. For example, the sentences beginning on lines 64, 73, 177, and 244.

Fig. 1: Use consistent y-axis across the 2 panels

Fig. 9: I was rather alarmed by the sharp uptick in the half catch scenario. Of course, this makes sense once I think about it, but it would be good to describe such a stark result in the text, even if it is expected.

Appendix plots: All references to the appendix plots are listed as appendix A rather than the correct appendix plot. These references should be fixed. It would also improve readability if there were a cap/no cap column in the appendix plots (I.e., fill the matrix of panels column-wise instead of row-wise).

All ecosystem indicator plots: The red line and green fill will be difficult for readers with colorblindness. I would recommend a gray fill instead.

Experimental design

The authors equate steepness with resilience. Steepness describes the density-dependence of the stock-recruit relationship. While this is related to resilience, particularly at low spawning biomasses, it is not correct to describe steepness as the resilience of the population across all biomasses. (I would also argue the term “resilience” really only applies at low biomasses anyway, but that is more debatable.) Other parameterizations for the Beverton-Holt model exist that could make more sense in this case—for example the asymptotic recruitment and slope at the origin. So while I agree that some of the dynamics of the uncapped recruitment model are odd, those that rely on the interpretation of steepness as resilience should be examined more closely. As the authors say, if the relationship must go through two fixed points (one being the origin) and has a low initial slope, it must by definition have a high slope later on.

Equation 2: Some rationale for this should be provided.

In order for a reader to replicate the results, the Atlantis model in McGregor et al. 2018 must be publicly available somewhere. Readers should be directed to a github repository or similar for the model.

Validity of the findings

The authors make a several a couple of poignant points in the discussion. First, that their different results for capped and uncapped recruitment do not necessarily imply which is correct (line 304, also paragraph on line 323). Second (and more importantly, I think), that how we parameterize multispecies models has a huge impact on what we infer the effects of predation release to be (line 319). This could even be emphasized!

Paragraph starting on line 248: It looks to me like the ratio exceeds the previously observed values in all recruitment scenarios, which is not how I interpreted what the authors wrote. Also, how do they know this behavior is “falsely” suggesting a change in ecosystem structure? Couldn’t it be correctly suggesting the change?

Fig. 3 and 4: I was curious how the population trajectories appear so smooth when the M time series is not smooth.

Additional comments

In this paper the authors describe a problem that results from ecosystem models borrowing parameterizations for population dynamics from single species models that are generally used on stochastic depleted populations. When these same population models are used in deterministic models on populations near their unfished biomasses, unusual dynamics where populations increase rapidly following predator release (or other declines in mortality) occur. The authors propose to cap recruitment at its unfished level rather than allowing it to increase towards an asymptote, and proceed to explore the impact of doing so on a variety of common indicators used in the analysis of ecosystem models.

Line edits:

44: While size-based models may be simpler, I think they are still aimed for more strategic uses, unlike MICE.

51: Beverton and Holt is not a 2012 paper.

314: Would be good to cite a paper that doesn’t critique balanced harvesting, too

Reviewer 2 ·

Basic reporting

The manuscript is understandably written and contains sufficient references. It does however contain minor grammatical errors, see for example line 132 and line 161-166.
Also, the reference McGregor et al (2019) should be updated.

Some figures would benefit from more detail. For example in Figure 4, an explanation regarding the relevance of the dashed line would be helpful.

The manuscript would improve by having terms like 'capped recruitment' (line 122) be explained, especially since it will determine model simulations (see line 170).

Experimental design

Overall, the method and data selection should include more details. To be more specific:

Section 2 would benefit from additional information:
The functional form of the Beverton Holt equation should be provided as it determines the value R. Further, more details should be added regarding the functional form for the time-dependent natural mortality, especially because it was mentioned in the section header. The expected value of M was provided but I suggest to add the underlying distribution as well as the choice of Variance etc. This will help the reader to follow the interpretations of Figure 2 and its implications.

Section 3:
Some data, that is being used in this paper, is not sufficiently introduced. For example, Figure 5 refers to catch data by McGregor et al (2019) but even in that publication the origin of the data is not revealed. This will however have implications. Often data is only available after data documentation commences. That does however not imply that there has not been predation earlier. This possibility should be addressed by the authors as well as the reliability of McGregor's data.
The overall model including fishing of predators should be presented as well as the choice of natural mortality to obtain the results illustrated in Figure 6. Although data from Atlantis was mentioned, it should be pointed out where and how it was incorporated in obtaining the results in Figure 6.

Section 4:
As the diversity measures require data from other species, it should be pointed out which data was used to obtain these values.
Although a simple model was introduced, line 225 mentions an age-structured model that seems to be explained in caption of Figure 8 but should be introduced whenever first needed.

Validity of the findings

The results indicate an interesting behaviour and build the base for further investigations.

Since simulations with 'capping' and 'non-capping' of recruitment seem to yield different results, it would be worth adding a paragraph in the Discussion regarding which choice of more realistic or how to determine which one would be.

---

## Round 0.2 · accepted · Accept

The authors addressed the reviewers' concerns and substantially improved the content of MS.

So, based on my own assessment as an editor, no further revisions are required and the MS can be accepted in its current form.